

# Decision support for the selection of reference sites using [137]Cs as soil erosion tracer

Laura Arata[1,*], Katrin Meusburger[1], Alexandra Bürge[1], Markus Zehringer[2], Michael E. Ketterer[3], Lionel Mabit[4] and Christine Alewell[1]

[1] Institute of Environmental Geosciences, Department of Environmental Sciences, University of Basel, Switzerland
[2] State Laboratory Basel-City, Basel, Switzerland
[3] Chemistry Department, Metropolitan State University of Denver, Colorado, USA
[4] Soil and Water Management & Crop Nutrition Laboratory, FAO/IAEA Agriculture & Biotechnology Laboratory, Austria.

*Correspondence to*: Laura Arata (laura.arata@unibas.ch)

**Abstract.** The classical approach to use [137]Cs as soil erosion tracer is based on the comparison between stable reference sites and sites affected by soil redistribution processes, and enables to derive soil erosion and deposition rates. The method is associated with potentially large sources of uncertainty with major parts of this uncertainty being associated with the selection of the reference sites. We propose a decision support tool to Check the Suitability of reference Sites (CheSS) for systematic validation or rejection of reference sites. The method is based on a repeated sampling approach, where the reference sites are resampled after a certain time period. Suitable reference sites are expected to present no significant temporal variation in their decay corrected [137]Cs depth profiles. Possible causes of temporal variation are narrowed down by a decision tree. More specifically, the decision tree tests for (i) uncertainty connected to small scale variability of [137]Cs due to its heterogeneous initial fallout (such as in areas affected by the Chernobyl fallout), (ii) signs of erosion/deposition processes, (iii) artefacts due to the collection, preparation and measurement of the samples and (iv) finally, if none of the above can be assigned, this variation might be attributed to "turbation" processes (e.g. bioturbation, cryoturbation and mechanical turbation such as avalanches or rock falls). CheSS has been tested in one Swiss alpine valley, where the suitability of six reference sites was tested.

## 1 Introduction

Soil erosion is a global threat (Lal, 2003). Recent estimated erosion rates range from low rates of 0.001–2 t ha[-1] yr[-1] on flat relatively undisturbed lands (Patric, 2002) to high rates under intensive agricultural use of > 50 t ha[-1] yr[-1]. In mountainous regions, rates ranging from 1–30 t ha[-1] yr[-1] have been reported (e.g. Descroix et al. 2003, Frankenberg et al. 1995, Konz et al., 2012) where they often exceed the natural process of soil formation (Alewell et al., 2015). The use of the artificial radionuclide [137]Cs as soil erosion tracer has been increasing during the last decades, and the method has been applied all over the world with success (e.g. Mabit et al., 2013; Zapata, 2002). The use of [137]Cs as soil erosion tracer allows an integrated



temporal estimate of the total net soil redistribution rate per year since the time of the main fallout, including all erosion processes by water, wind and snow during summer and winter seasons (Meusburger et al., 2014).

[137]Cs was released in the atmosphere during nuclear bomb tests and as a consequence of nuclear power plant (NPP) accidents such as Chernobyl in April 1986. It reached the land surface by dry and wet fallouts and once deposited on the ground, it is

strongly bound to fine particles at the soil surface. Due to its low vertical migration rates, it moves predominantly in association with fine soil particles through physical processes, and provides an effective track of soil and sediment redistribution processes (Mabit et al, 2008). The traditional approach in using the [137]Cs method is based on the comparison between the inventory (total radionuclide activity per unit area) at a given sampling site and that of a so-called reference site, located in a flat and undisturbed/stable area. The method indicates the occurrence of erosion processes at sites with lower

[137]Cs inventory as compared to the reference site, and sediment deposition processes at sites with a greater [137]Cs inventory (Figure 1, A). Specific mathematical conversion models allow then to derive from the latter comparison quantitative estimates of soil erosion and deposition rates (IAEA, 2014).

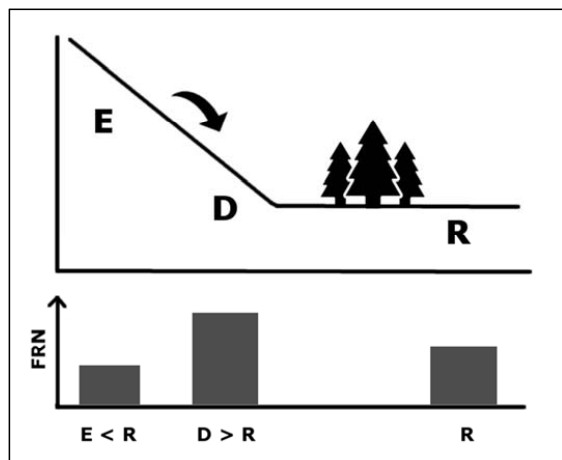

**Figure 1: Concept of the fallout radionuclide (FRN) traditional method, in which the FRN content of a reference site located in a**
**flat and undisturbed area (R) is compared to the FRN content of disturbed sites (E and D). If the FRN at the site under investigation is lower than at the reference site, the site has experienced erosion processes (E), while if the FRN content is greater than at the reference site, the site has experienced deposition processes (D).**

The efficacy of the method relies on an accurate selection of representative reference sites (Mabit et al., 2008; Owens and Walling, 1996, Sutherland,1996). The measured total [137]Cs inventory at the reference sites represents the baseline fallout (i.e.

reference inventory), a fundamental parameter for the qualitative and quantitative assessment of soil redistribution rates (Loughran et al., 2002). It is used for the comparison with the total [137]Cs inventories of the sampling sites, and therefore determines if and how strongly a site is eroding or accumulating sediments. Moreover, the depth profile of the [137]Cs distribution in the soil at the reference site plays a very important role, as the shape of this profile is used in the models to



convert changes in [137]Cs inventory changes to quantitative estimates of soil erosion rates (Walling et al., 2002). Recent studies demonstrated the sensitivity of conversion models to uncertainties or even biases in the reference inventory (e.g. Arata et al., 2016; Iurian et al., 2014; Kirchner, 2013).

A close proximity of a reference site to the area under investigation is required to meet the assumption that both experienced

similar initial fallout. The latter is particularly important if the study area was strongly affected by Chernobyl fallout, which is, besides global fallout from nuclear weapons testing, the major input of [137]Cs in many regions of Europe. Because of different geographical situations and meteorological conditions at the time of passage of the radioactive cloud, the contamination associated with Chernobyl fallout was very inhomogeneous (Chawla et al., 2010, Alewell et al., 2014) . Therefore, in some areas a significant small scale variability of [137]Cs distribution may be expected and, as already pointed

out by Lettner et al., (1999) and Owens and Walling (1996), might impede the comparison between reference and sampling sites. To consider adequately the spatial variability of the FRN fallout, multiple reference sites should be selected and the variability within the sites properly tackled (Kirchner, 2013, Mabit et al., 2013, Pennock and Appleby, 2002). In addition, the reference site should not have experienced any soil erosion or deposition processes since the main [137]Cs fallout (which generally requires that it was under continuous vegetation cover such as perennial grass). Different forms of turbation,

including animal- or anthropogenic impact and cryoturbation or snow processes may also affect the [137]Cs soil depth distribution at the reference site. Finally, the collection of the samples, the preparation process and the gamma analysis might introduce a certain level of uncertainty, which should be carefully considered. For instance, Lettner et al. (1999) estimated that the preparation and measuring processes contribute 12.2% to the overall variability of the reference inventory. Guidance in form of independent indicators (e.g. stable isotopes as suggested by Meusburger et al., 2013) for the suitability of

reference sites might help to assist with the selection of reference sites.

We propose an alternative method to Check the Suitability of reference Sites (CheSS) using a repeated sampling strategy. Our basic assumption is that decay corrected [137]Cs depth profiles measured in two points in time should be identical. The suitability of reference sites for an accurate application of [137]Cs as soil erosion tracer is tested at at Urseren Valley (Canton Uri, Swiss Central Alps).

**2 CheSS (Check the Suitability of reference Sites): a concept to assess the suitability of reference sites for proper application of [137]Cs as soil erosion tracer**

### 2.1 Repeated sampling strategy and calculation of inventories

The time period for the repeated sampling of reference sites needed for the application of [137]Cs as soil erosion tracer will be case specific and depends on the initial small scale spatial variability of the reference inventory. Several spatial repetitions

are necessary and should be analyzed separately to investigate the small scale variability of [137]Cs in the area. As we detected measurement differences between different detectors (see below), all samples should ideally be measured for [137]Cs activity





using the same analytical facilities. Finally, $^{137}$Cs activity needs to be decay corrected to the same date (either the period of the first sampling campaign or the second one), considering the half-life of $^{137}$Cs (30.17 years).

The decay corrected $^{137}$Cs activities (*act*, Bq kg$^{-1}$), of each soil layer of the depth profile are converted into inventories (*inv*, Bq m$^{-2}$) with the following equation:

$Inv = act \times xm$                                                                            (1)

where *xm* is the measured mass depth of fine soil material (<2 mm fraction) (kg m$^{-2}$) of the respective soil sample. The depth profile of each reference site is then displayed as inventory (Bq m$^{-2}$) against the depth of each layer (cm). The repeated-sampling inventory change (*Inv$_{change}$*) can then be defined as:

$$Inv_{change} = \frac{Inv_{t0} - Inv_{t1}}{Inv_{t0}} \times 100$$                                      (2)

where $t_0$ and $t_1$ are the dates of the first and the second sampling campaigns respectively, $Inv_{t1}$ is the $^{137}$Cs inventory (Bq m$^{-2}$) at $t_1$, and $Inv_{t0}$ is the $^{137}$Cs inventory at $t0$. Positive values of $Inv_{change}$ indicate erosion, whereas negative values stand for deposition.

**2.2 A decision tree to identify possible pitfalls for the suitability of reference sites**

Our aim is to assess the suitability of the reference sites by analyzing for a possible temporal variation of the $^{137}$Cs inventory.
Given the assumption that no additional deposition of $^{137}$Cs occurred at the sites during the investigated time window (which is valid worldwide except for the areas affected by the Fukushima-Daiichi fallout), any temporal variation of the $^{137}$Cs content should be attributable to different forms of soil disturbance or to artefacts in the preparation/measurement of the samples. The potential causes of the temporal variation in the $^{137}$Cs total inventories and depth profiles are examined through a decision tree which includes six main nodes (Figure 2).

**Node 1: No significant temporal variation of the $^{137}$Cs total inventory**

Firstly, the temporal variation of the $^{137}$Cs total inventory at each reference site is tested. Ideally, in both sampling campaigns several replicates have been collected. Then a suitable test for significant differences should confirm or reject the hypothesis of $^{137}$Cs total inventory stability. If differences between the sampling years are significant, then the site should not be considered suitable for further application. Otherwise, if no significant differences are found, then it is possible to pass to the
second node of the decision tree to test for significant differences in the depth profiles.

However, in some cases at one of the sampling campaigns (most likely the first one, at $t_0$) no replicates have been collected, or composite soil samples have been collected, and therefore it is not possible to run statistical tests. To overcome this situation, the following approach is proposed. First, the spatial variability among the replicates collected for a single



campaign is assessed by the coefficient of variation (e.g. $CV_{t1}$). As $^{137}$Cs should be considered like any other soil property, a moderate spatial variability is to be expected, characterized by a CV of 15-35% (see Sutherland, 1996).

If the $CV_{t1}$ exceeds the suggested range, this could be a sign of a heterogeneous distribution during the deposition of the initial $^{137}$Cs fallout. Given that, the site should not be considered suitable, and the analysis could directly pass to the third

node of the decision tree. If the $CV_{t1}$ is still within the valid range, then a new cumulative CV is calculated ($CV_{t0+t1}$), which expresses the variation among the replicates collected at t0 and t1. If the $CV_{t0+t1}$ still does not exceed the range proposed by Sutherland, it is possible to pass to the second node of the decision tree, where the analysis focuses on the temporal variation of the depth profile. If the $CV_{t0-t1}$ exceeds the range, then again the site should be excluded.

**Node 2: No significant temporal variation of the $^{137}$Cs depth profile**

Secondly, it is tested whether there is a significant temporal variation between the $^{137}$Cs depth profiles measured in *t0* and *t1*. In theory, the shape of the depth profile should not have changed between t0 and t1 if the site remained undisturbed. If replicate samples have been collected during the two sampling campaigns, a statistical test (for example the t-test or the Wilcoxon test) is run for the $^{137}$Cs inventories of each depth increment between the groups of replicate samples collected in *t0* and *t1* (Figure 2, node 2). If there is a significant difference between the shapes of the two depth profiles, the site should

not be considered appropriate for the $^{137}$Cs method application. If no replicates have been collected, or if samples have been mixed, it is necessary to follow the same approach proposed in section 3.2. The CV among the same depth increment within the replicates collected at one sampling campaign is analyzed. Again, it is possible to consider the CV range reported by Sutherland (1991) as a threshold value for the determination of the suitability of the site. If the CV exceeds that range, the site should not be used, and the causes of the temporal variation could be examined in the subsequent nodes of the decision

tree. Otherwise, if there is no significant change in the depth profile between *t0* and *t1*, the single value measured during the second sampling campaign is added, and the change in the CV is investigated. If the final CV is lower than the CV range proposed by Sutherland, the site can be further considered valid for the application of the $^{137}$Cs method. In any other cases, a deeper analysis through the decision tree is needed.




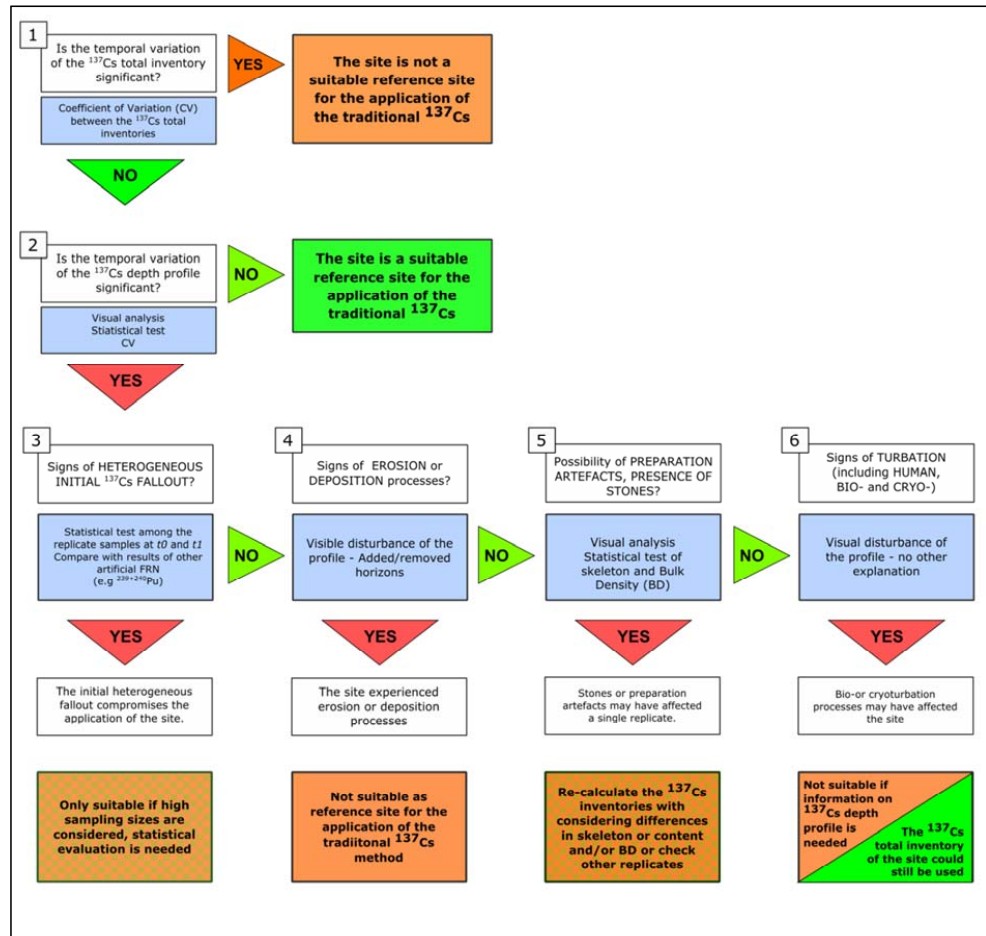

**Figure 2: The CheSS decision tree to validate the suitability of a reference site for using $^{137}$Cs as soil erosion tracer.**

**Node 3: Signs of a heterogeneous initial fallout of $^{137}$Cs over the area**

A significant difference between the reference $^{137}$Cs depth profiles measured at *t0* and at *t1* may not be necessarily due to a
5   temporal variation because of soil disturbance, but instead be caused by a high small scale spatial variability of $^{137}$Cs
distribution at the site, due to heterogeneous initial fallout over the study area (Figure 2, node 3). In Europe, significant small
scale variability of $^{137}$Cs distribution is known to be due to the Chernobyl fallout, which was characterized by a high $^{137}$Cs
deposition associated with few rain events. Compared to the nuclear bomb tests fallout, the Chernobyl fallout was



significantly more heterogeneous (e.g. Alewell *et al.*, 2014). Therefore, in the areas affected by the Chernobyl fallout, sites sampled closely to each other may present very different $^{137}$Cs contents. It is therefore necessary to investigate the small scale spatial variability (e.g. the same scale as distance between reference site replicates) measured at both or at least one sampling campaign, looking at the CV again, as presented in the previous sections, or through a statistical test (for example

the Analysis of the Variance, ANOVA). If the spatial variability is highly significant, the site should not be envisaged as a reference site for the application of the $^{137}$Cs method unless the number of samples collected for the determination of the reference baseline is large enough (at least 10) to counterweight the small scale variability within the site (Mabit *et al.*, 2012; Sutherland, 1996, Kirchner,2013). A possible validation of this cause of heterogeneity might be a comparison with the spatial distribution of another FRN such as $^{239+240}$Pu. (Figure 2, node 3). As the fallout deposition of $^{239+240}$Pu after the

Chernobyl accident was confined to a restricted area in the vicinity of the Nuclear Power Plant (Ketterer *et al.*, 2004), the origin of Plutonium fallout in the rest of Europe is linked to the past nuclear bomb tests only. Consequently, Pu fallout distribution was more homogeneous (Alewell *et al.*, 2014; Ketterer *et al.*, 2004; Zollinger *et al.*, 2015). If the $^{239+240}$Pu depth profiles do not vary significantly between the two sampling years, there should be no disturbance (e.g. turbation, erosion) or measurement artefacts. As such, it might be concluded that the heterogeneous deposition of $^{137}$Cs at the time of the fallout

prejudices the use of Cs at this site.

**Node 4: Signs of disturbance associated with erosion and deposition processes**

The temporal variation in the $^{137}$Cs depth profile may have been caused by soil movement processes affecting the site during the two sampling periods (Figure 2, node 4). If the site experienced a loss of soil due to erosion, we expect to observe a removal of the top soil layers of the profile measured during the second sampling campaign (Figure 3, A). In case of

deposition, a sedimentation layer should be found on the top of the reference depth profile, assuming that no ploughing operations affected the site (Figure 3, B). If information on the depth distribution of another FRN is available, this might provide a reliable confirmation.




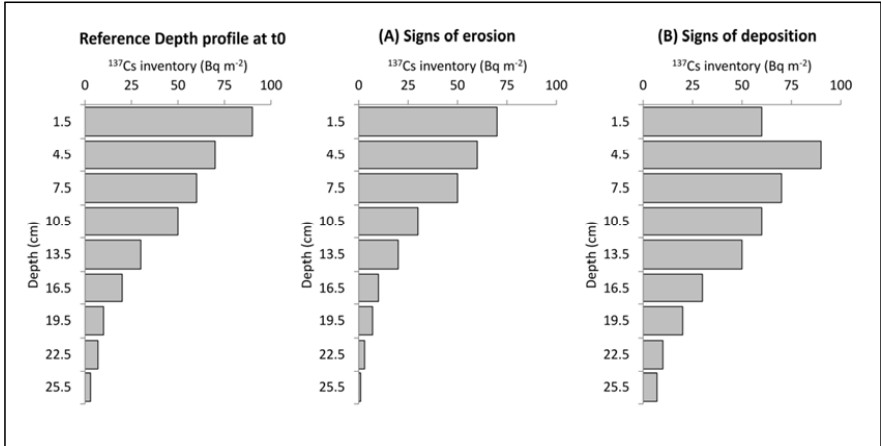

**Figure 3: Signs of sheet erosion (A) and deposition (B) on a depth profile of a hypothetical reference site at time 1.**

**Node 5: Sampling or preparation artefacts**

One very common artefact which might bias the comparison between the samples collected at *t0* and *t1* is the difference in

5 the skeleton content (the percentage of soil fractions > 2mm) (Figure 2, node 5). The presence of stones might determine pass ways of water as well as very fine particles and solutes in the soil and thus influences the accumulation/migration of $^{137}$Cs through the soil layers. As $^{137}$Cs reaches the soil by fallout from the atmosphere, the common shape of the $^{137}$Cs distribution along the undisturbed depth profile can be described by an exponential function, with the highest $^{137}$Cs concentrations located in the uppermost soil layers (Mabit *et al.*, 2008; Walling *et al.*, 2002). This is particularly the case for

10 soils with low skeleton content (Figure 4, A) since the presence of stones may affect $^{137}$Cs depth distribution either through (i) impeding the $^{137}$Cs downward migration ($^{137}$Cs activity could then be concentrated in the layer above the stone (Figure 4, B) or (ii) creating macro- and micro-pores favouring $^{137}$Cs associated with fine particles to "migrate" to deeper layers (Figure 4, C) or causing lateral movement which will induce a lower $^{137}$Cs content in our samples.





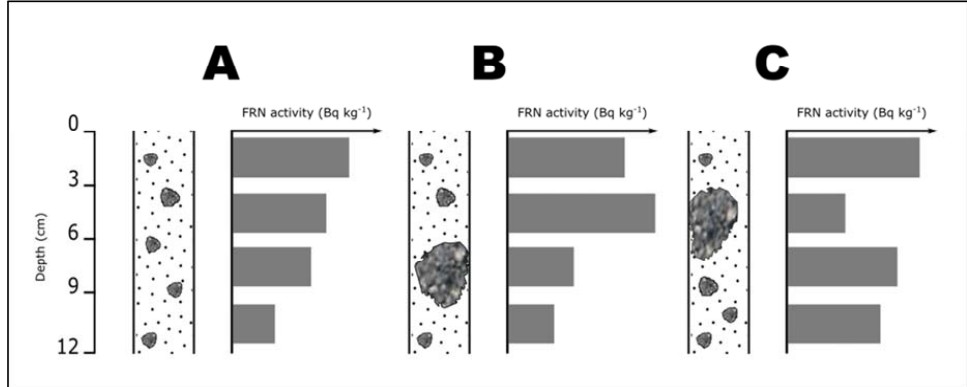

**Figure 4: Possible influence of stones on the FRN depth distribution.**

As such, the seemingly temporal variation in the depth profile might indeed be a spatial variation induced by differences in skeleton content and/or bulk densities. Higher bulk densities will result in higher increment inventories even if $^{137}$Cs

activities at the layers are comparable. Thus, a thorough control (eventually through a statistical test such as paired T test) if skeleton content and bulk densities are comparable between the sampling years is highly suggested. Finally, sampling, preparation artefacts and measuring processes may produce various sources of errors between the years. The latter is especially the case, if different people prepared the samples between the years. An estimation of possible errors might be considered, for example through a simulation of different increment assignment along the profile. If different detectors or

different calibration sources and/or geometry are used in the two sampling campaigns, a comparability check of the measurements is advisable. For instance, a subset of samples could be measured with the two different detectors and any potential discrepancy of the results should be properly reported.

**Node 6: Signs of soil disturbance**

Different forms of disturbance, such as bio-, cryoturbation or even human induced soil perturbation (e.g. tillage, seed bed

preparation etc.) might have influenced the $^{137}$Cs depth distribution between *t0* and *t1* (Figure 2, node 6). Occurrences of turbation are often difficult to identify prior to sampling but might eventually be detected by using other tracing approaches, such as the $\delta^{13}$C depth distribution (Meusburger *et al.*, 2013; Schaub and Alewell, 2009). If every other possible cause (node 3 to 5 of the decision tree) has been excluded, then its $^{137}$Cs depth distribution should not be considered in the estimation of soil redistribution rates. Nonetheless, the total inventory of $^{137}$Cs at the site could still be used. Indeed, there are simple and

basic mathematical conversion models, such as the proportional model (Ritchie and McHenry, 1990, IAEA, 2014), which require information only about the total reference inventory of $^{137}$Cs, and do not need detailed information about the $^{137}$Cs depth distribution.



### 3 The application of the CheSS decision tree

#### 3.1 Study area

To test the methodology described above, we used a dataset from an alpine study area, the Urseren Valley (30 km$^2$) in Central Switzerland (Canton Uri), which has an elevation ranging from 1440 to 3200 m a.s.l. At the valley bottom (1442 m
a.s.l.), average annual air temperature for the years 1980–2012 is around 4.1 ± 0.7 °C and the mean annual precipitation is 1457 ± 290 mm, with 30% falling as snow (MeteoSwiss, 2013). The U-formed valley is snow-covered from November to April. On the slopes, pasture is the dominant land use, whereas hayfields are prevalent near the valley bottom.

#### 3.2 Sampling design

Supportive information was provided by the local landowners to select the reference sites in both valleys. Sites used for
ploughing and grazing activities were excluded. A first sampling campaign was undertaken in autumn 2013. Six reference sites were identified in flat and undisturbed areas along the valley. At each site 3 cores (40 cm depth), 1 m apart from each other, were sampled. The cores were cut in 3 cm increments, to derive information on the $^{137}$Cs depth profile. The three cores from each site were bulked to provide one composite sample per site. During spring 2015, all six reference sites were resampled. Considering the typical and high redistribution dynamics of the valley, the time span is sufficiently long to ensure
the possibility to observe changes in the depth profiles if soil erosion and deposition processes affected the area. At each site, we collected three replicates, which were analyzed separately, to investigate the small scale variability of the FRN content. All cores were air-dried (40°C for 72h), sieved (<2 mm) to remove coarse particles and the skeleton content as well as the bulk density (BD) was determined.

#### 3.3 Measurement of anthropogenic FRN activities and inventories

The measurements of the $^{137}$Cs activity (Bq kg$^{-1}$) were performed with high resolution HPGe detectors. The $^{137}$Cs activity (Bq kg$^{-1}$) from 2013 were analysed at the Institute of Physics of the University of Basel using a coaxial, high resolution germanium lithium detector (Princeton Gammatech) with a relative efficiency of 19% (at 1.33 MeV, $^{60}$Co). Counting time was set to 24 hours per sample. Samples collected in 2015 were analysed at the state laboratory Basel-City using coaxial high resolution germanium detectors having 25% to 50% relative efficiencies (at 1.33 MeV, $^{60}$Co). Counting times were set
to provide a precision of less than ±10% for $^{137}$Cs at the 95% level of confidence.

All soil samples were counted in sealed discs (65 mm diameter, 12 mm height, 32 cm$^3$) and the measurements were corrected for sample density and potential radioactivity background. The detectors located at the state laboratory Basel-City were calibrated with a reference solution of the same geometry. The reference contained $^{152}$Eu and $^{241}$Am (2.6 kBq rsp. 7.7 kBq) to calibrate the detectors from 60 to 1765 keV. It was obtained from the Czech Metrology Institute, Prag. This solution
was bound in silicon resin of a density of 1.0. The efficiency functions were corrected for coincidence summing of the $^{152}$Eu lines using a Monte Carlo simulation program (Gespecor). The $^{137}$Cs was counted at 662 keV with an emission probability of




0.85 and a (detector) resolution of 1.3 to 1.6 keV (FWHM). All measurements and calculations were performed with the gamma software Interwinner 7. The [137]Cs activity measurements were all decay corrected to the year 2015.

To compare the [137]Cs results to another artificial FRN, all samples were also measured for [239+240]Pu activity. The determination of Plutonium isotopes from both valleys and for both sampling years were performed using a Thermo X Series

5   II quadrupole ICP-MS at the Northern Arizona University, USA. Detailed description of the ICP-MS specifications and sample preparation procedure can be found in Alewell *et al.*, 2014. The activities of [137]Cs and [239+240]Pu (*act*, Bq kg$^{-1}$) were converted into inventories (Bq m$^{-2}$) according to equation (1).

### 3.4 Application of the CheSS decision support tool to the reference sites

Because the [137]Cs activity of the samples was measured with different detectors for the two sampling years, we investigated

10   the potential variability between the two detectors. A selected subset of samples (n= 24) was analysed using both detectors (i.e. the one located at the Institute of Physics of the University of Basel and the other located at the State Laboratory Basel-City). The results highlight a high correspondence of the measurements held by the two analytical systems ($R^2 = 0.97$; $p <$ 0.005), however the detector of the State Laboratory Basel-City returns slightly lower [137]Cs activities (Figure 5). Thus, the [137]Cs activity of the samples measured in 2013 was corrected to the values of the detector of State Laboratory Basel-City

15   (higher efficiency) to allow comparability between the different data sets.

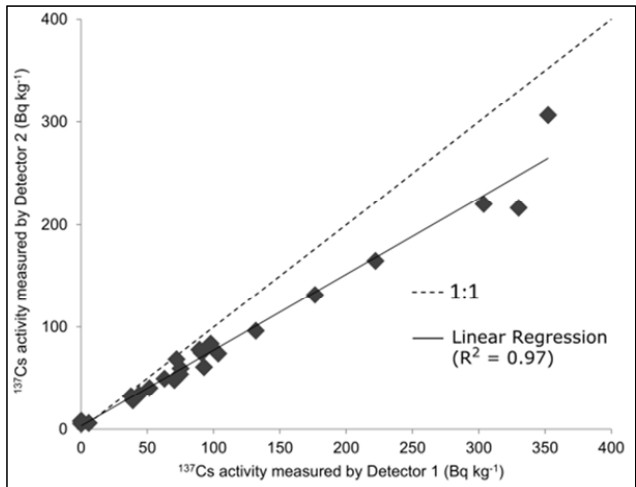

**Figure 5: The comparison between the [137]Cs measurements of a subset of samples (n=16) performed with two different HpGe detectors, where detector 1: detector hosted at the Physics department of the University of Basel (CH) and detector 2: detector hosted at the State-Laboratory of Basel (CH) .**



Total $^{137}$Cs inventories (decay corrected to the year 2015) of the six reference sites collected in the Urseren Valley in 2013 range from 3858 to 5057 Bq m$^{-2}$, with a mean value of 4515 Bq m$^{-2}$ and a standard deviation (SD) of 468 Bq m$^{-2}$. Data from 2015 range between 3925 to 8619 Bq m$^{-2}$, with a mean value of 5701 Bq m$^{-2}$ and a SD of 1730 Bq m$^{-2}$ (Figure 6).

All reference sites have been investigated through the CheSS decision tree presented in section 2.2. As for the first node, we investigated the temporal variation in the $^{137}$Cs total inventories at each reference site. The replicate samples were analyzed separately only during the second sampling campaign, while during the first sampling campaign only bulked samples were analysed. Thus, the approach presented in section 3.2 (Node 1) was followed, and the analysis first focused on the spatial variability measured at t1. Reference sites 3, 5 and 6 presented signs of high small scale variability, as expressed by CV of 48 % for all cases (Table 1). Such variability excluded them from any further application as reference sites. As for reference sites 1, 2 and 4, the CV$_{t0-t1}$, calculated among the three replicates plus the single replicate measured at t1, was then investigated. At reference site 4, the CV$_{t0-t1}$ reached a value of 41%, and indicated a significant temporal variation between the two sampling years. The site should also be excluded. As for reference sites 1 and 2, their suitability was checked in the next node of the decision tree.

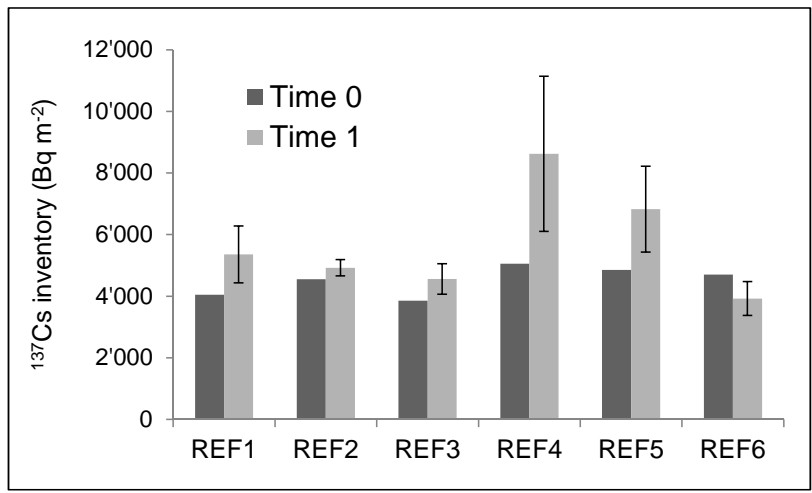

**Figure 6: Temporal variation between the total $^{137}$Cs inventories measured at the reference sites in the Urseren Valley, where Time 0 = 2013 and Time 1 = 2015. The errors bars indicate the standard deviations of the inventories among the replicates collected at each reference in 2015.**

Passing at the second node of the CheSS decision tree presented in section 2, the analysis focuses on the temporal variation of the $^{137}$Cs depth profile (figure 7). Also for this node, we examined CV$_{t1}$ which in this case expresses the variation among each depth increment (Table 2). For both sites under investigation (i.e. reference site 1 and reference site 2) the CV$_{t1}$ values



exceed the range suggested by Sutherland in most layers. Thus, the two sites could not be considered suitable for further application.

**Table 1: The Coefficients of Variation measured for the $^{137}$Cs total inventory at the reference sites, where $CV_{t1}$ is calculated among the three replicates collected at *t1* and $CV_{to-t1}$ is calculated among the replicates collected at t1 and the single replicate measured at**

5      **t0.**

|      | $CV_{t1}$ | $Cv_{to-t1}$ |
|------|------|------|
| REF1 | 19   | 33   |
| REF2 | 26   | 28   |
| REF3 | 48   | 48   |
| REF4 | 31   | 43   |
| REF5 | 48   | 52   |
| REF6 | 48   | 41   |





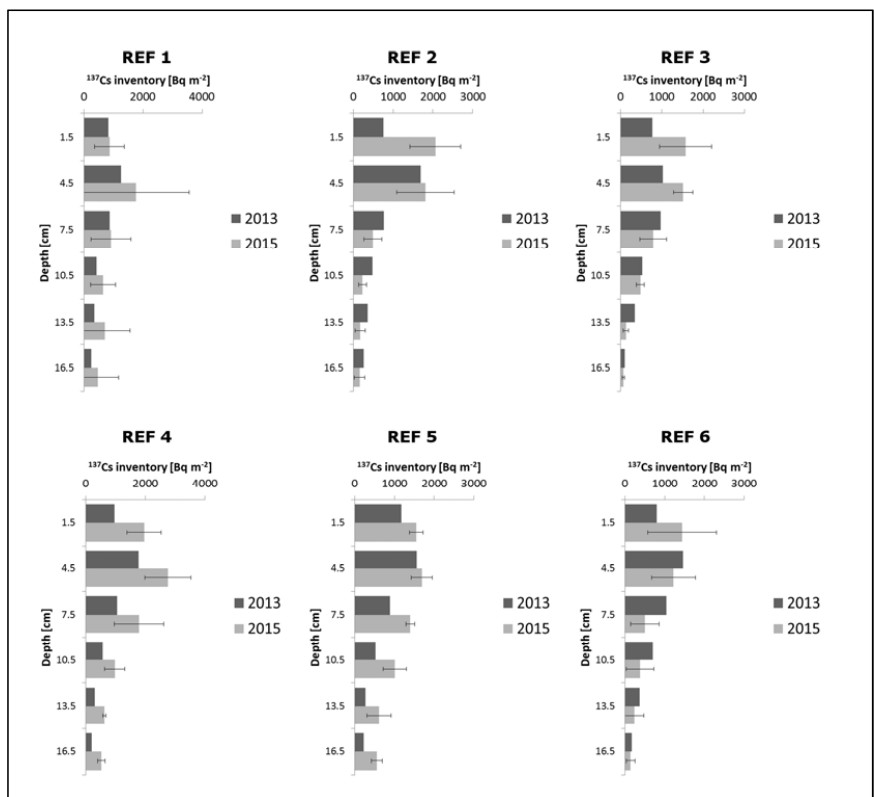

**Figure 7: The $^{137}$Cs depth profiles of the nine investigated reference sites in the Urseren Valley for the two different sampling campaigns. The errors bars indicate the standard deviations of the inventories among the replicates collected at each reference site 2015.**

5    Moreover, an Analysis of the Variance (ANOVA) test revealed that the replicates collected at reference site 2 differ significantly from each other (p-value of 0.034), confirming a small scale variability within the site. To validate those results, $^{239+240}$Pu inventories measured at the three replicates of the site were also analysed (Figure 8). In this case, the test highlighted no significant difference between the replicates. Thus, at reference site 2, the heterogeneous distribution of $^{137}$Cs might be due to the heterogeneous Chernobyl deposition on snow covered ground. In contrast, the $^{239+240}$Pu depth profiles of

10    the three replicates at reference site 1 present also significant differences. We then looked at the differences in the skeleton content of the three replicates. An ANOVA test showed a significant difference (p-value of 0.025), thus, a difference in the presence of stones in the three soil cores might affect the FRN depth distribution. In particular a Tukey's HSD (Honest Significant Difference) Post-hoc pairwise comparison identified the replicate number 3 as a potential outlier. To validate the



suitability of the site more replicates should be collected and measured, in order to compare their [137]Cs depth profiles to the results obtained during the first sampling campaign.

**Table 2: The Coefficients of Variation (%) between the [137]Cs inventory of each depth increment at the reference sites measured at *t0* and *t1*.**

| Depth | REF1 | | REF2 | |
|---|---|---|---|---|
| | $CV_{t1}$ | $Cv_{to-t1}$ | $CV_{t1}$ | $Cv_{to-t1}$ |
| 1.5 | 57 | 47 | 31 | 48 |
| 4.5 | 97 | 88 | 41 | 34 |
| 7.5 | 73 | 61 | 45 | 40 |
| 10.5 | 69 | 63 | 42 | 50 |
| 13.5 | 120 | 121 | 72 | 62 |
| 16.5 | 140 | 136 | 82 | 64 |




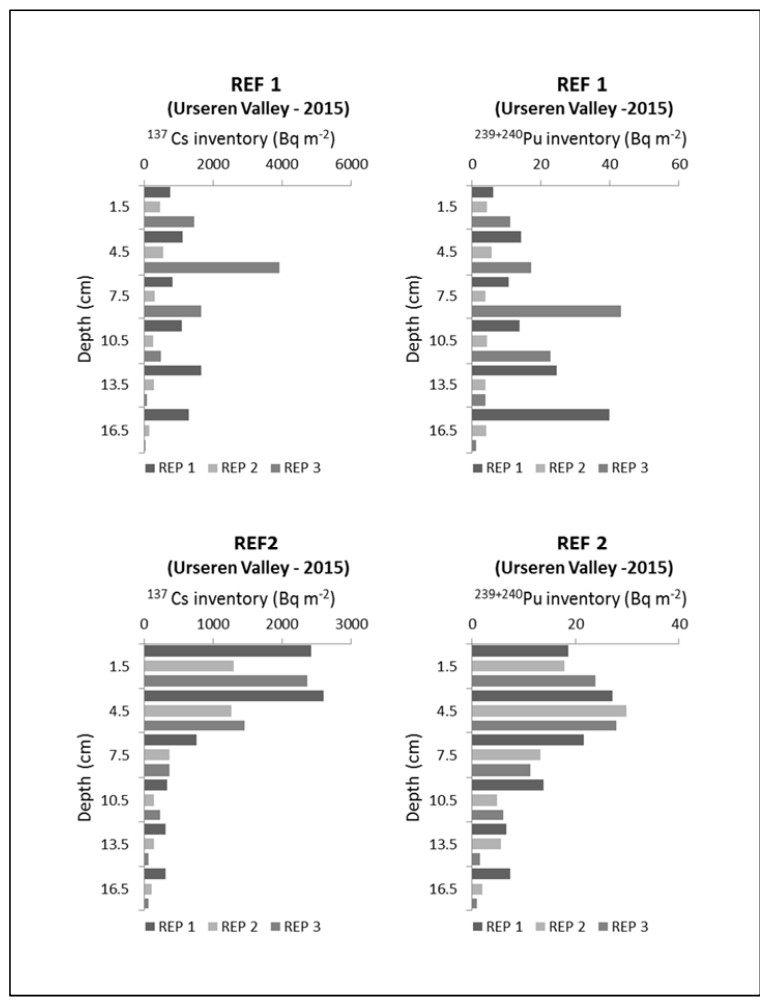

**Figure 8: The** $^{137}$**Cs depth profiles (on the left) and the** $^{239+240}$**Pu depth profiles (on the right) of the three replicates collected at reference sites 1 and 2 in 2015 in the Urseren Valley.**

#### 4 Conclusion

5    With the decision tree CheSS, the authors developed a support tool to verify the suitability of reference sites for a $^{137}$Cs based soil erosion assessment. Great attention has to be given to the analysis of the small scale variability of $^{137}$Cs



distribution in the reference areas, especially in those regions affected by Chernobyl fallout. To cope with small scale variability, sampling numbers might be increased or the temporal variation of another artificial radionuclide, such as $^{239+240}$Pu might be analysed. The CheSS test in the Urseren Valley indicated that the heterogeneity of $^{137}$Cs distribution prejudiced the suitability of most reference sites. At reference site 1 the presence of stones affected the shapes of the depth

profile in at least one replicate sample. Therefore, the application of the traditional $^{137}$Cs approach, based on a spatial comparison between reference and sampling sites, is compromised. To derive soil redistribution rates, a $^{137}$Cs repeated sampling approach should be preferred. This approach is based on a temporal comparison of the FRN inventories measured at the same site in different times (Kachanoski & de Jong, 1984). It doesn't require the selection of reference sites, because the inventory documented by the initial sampling campaign is used as the reference inventory for that point (Porto *et al.*,

10 2014).

Accurate soil erosion assessment is crucially needed to validate soil erosion modelling, which can help prevent and mitigate soil losses on a global scale. In this context, the $^{137}$Cs method could play a decisive role, if we are able to overcome its potential pitfalls, especially related to the selection of suitable reference site. The decision tree CheSS is a tool for objective and comparable testing, which enables to exclude those sites which present signs of uncertainty. With this we are convinced

to contribute improving the reliability of the $^{137}$Cs based soil erosion assessments.

**Authors contributions.**

L. Arata, K. Meusburger, L. Mabit and C. Alewell designed the concept of the method and analysed the data. A. Bürge contributed to the collection and preparation of the soil samples, and to the analysis of the data. M. Zehringer measured the $^{137}$Cs activity of the soil samples and analysed the results. M. E. Ketterer measured the $^{239+240}$Pu activity of the soil samples.

L. Arata prepared the manuscript with contributions from all co-authors.

**Acknowledgements**

The authors would like to thank Annette Ramp, Gregor Juretzko, Simon Tresch, Carmelo La Spada and Axel Birkholz for support during field work. This work was financially supported by the Swiss National Science Foundation (SNF), project no. 200021-146018, and has been finalized in the framework of the IAEA Coordinated Research Project (CRP) on "*Nuclear*

*techniques for a better understanding of the impact of climate change on soil erosion in upland agro-ecosystems*" (D1.50.17).



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
