# Peer review of "Decision support for the selection of reference sites using $^{137}\mathrm{Cs}$ as soil erosion tracer"

_SOIL, 2016_

## Referee Comment (RC1) · Anonymous Referee #1 · 25 Dec 2016

In this paper the authors explored an important issue in the application of the 137Cs technique that relates to the choice of a reference site. They proposed a decision support tool (CheSS) to check the suitability of a reference site using repeated measurements of 137Cs undertaken in 2013 and 2015 on the same sites and measurements of 239+240Pu carried out in 2015. The basic assumption is that suitable reference sites are expected to present no significant temporal variation in their decay corrected 137Cs depth profiles. The authors individuated four main causes of possible variation in the inventory. These are (1) small scale variability connected to the non-homogeneus fallout (see areas affected by Chernobyl), (2) signs of erosion and/or deposition, (3) artefacts due to sampling and measurements, (4) turbation processes. The authors screened their six reference sites in the Urseren Valley (Switzerland) based on this assumption and tried to individuate the suitable reference site. The paper seems to me very interesting and I think it should be published but some more details need to be added and/or discussed in this version.

Specific comments Introduction – The authors explain briefly the basic assumptions of the 137Cs with the help of Fig. 1. I agree with the explainations reported in the text but I think Figure 1 is a little bit misleading for people that are not familiar with the technique. In fact, what they depict as 'reference site (R)' is a valley bottom and, as is, it can be a depositional site. Also, what they depict as 'depositional site (D)' is a foot-slope and it may not necessarily be a 'depression' where deposition occurs. I suggest to redraw this figure in a more proper way (see my example below)

Fig. 1 (modified)

2.1 Repeated sampling strategy and calculation of inventories This is a very important point. More work must be done to establish how long the time period between two sampling campaigns should be. This depends on the 137Cs inventory of the reference site and on its spatial variability. If the inventory is low, it is difficult to understand if the difference between the two sampling campaigns should be attributed to decay or to erosion and/or deposition or to the detector efficiency. In this case, a period of at least 10-15 years could be necessary. If the inventory is high and it is affected by the Chernobyl fallout, we could expect that the small scale spatial variability and the temporal variability are of the same order of magnitude and it is difficult to distinguish the relative contributions. Something like that is suggested by the data that the authors show in their Fig. 6. The inventory provided in 2015 are all higher than those obtained two years before (in 2013) with the only exception of Ref 6. This is not unexpected because it is not possible to relocate exactly the same sampling points. Clearly, more samples are necessary in this case. But also, a time period of two years between two sampling campaigns could not be enough. The authors can add some comments here.

Node 2: No significant temporal variation of the 137Cs depth profile I agree with the test related to the total inventory as explained in Node 1. However, I found node 2 too

severe. I agree that the shape of the reference profile is important but, I think the test should be done on the entire profile not on the single layers. In many years of experience, I have never seen two profiles collected in the same site being identical. Maybe a practicle example can clarify my thoughts. Below there are 3 potential reference profiles characterised by the same total inventory (2510 Bq m-2), so they passed Node 1. They can be fitted by the same exponential model (same shape parameter h0 = 70 kg m-2 and same surface concentration A0 = 35 Bq kg-1). The values of cesium activity and mass depth for each layer are reported below.

Figure (My example)

Using the Sutherland range as a test (see suggestions in Node 2), in 5 cases out of 10 (see my values in red) the CV is greater than 35% (which is the upper limit suggested by Sutherland). If I have understood well, this result would suggest that the site where these profiles have been obtained is not suitable as reference site. I do not think I can agree with that because they show the same exponential decline with depth, and the difference between each single layer can be attributed to other factors (the authors mentioned some of the other causes later in the paper). On the contrary, if we use a t-test or other statistical tests to compare mean and variance of these three profiles, I may be wrong, but I did not find any statistical difference. I think the authors should think about it and add some comments.

There is another limitation in the application of this procedure suggested by the authors. In my example I have considered the same value of mass depth increment for the 3 profiles. This is an ideal case. In reality, due to differences in soil type, land use, presence of stones etc., it is difficult to obtain equal values of mass depth for the corresponding layers of different profiles. This makes this comparison not possible. In the end, I find more useful to check the shape of the entire profile.

Node 4 - Signs of disturbance associated with erosion and deposition processes I agree with the explaination in the text but, Figure 3 shows only the case where the

137Cs profile is perfectly exponential. In many places, profiles obtained in reference sites show a peak below the surface due to migration processes downward. In this case, the profile shown in figure 3b can be an undisturbed reference profile and, consequently, sheet erosion and deposition processes modify the shape accordingly. I suggest to improve their figure 3 considering both the possible cases (see my example below).

Fig. 3 (modified)

Page 7 – Line 21-22 – The authors say 'If information on the depth distribution of another FRN is available, this might provide a reliable confirmation'. I agree with this statement, but an example is necessary. The use of 210Pbex proved to be very effective in combination with or in alternative to 137Cs. In fact, good relationships exist between the results obtained with 137Cs and 210Pbex. I am sure the authors want to add some comments here maybe recalling some of the works done in this field (see for example Porto et al., 2006; 2013).

Porto P., Walling D.E., Callegari G. and Catona F. (2006). Using fallout lead-210 measurements to estimate soil erosion in three small catchments in Southern Italy. Water, air and soil pollution: focus 6, 657-667 Porto P., Walling D.E., Callegari G. (2013). Using 137Cs and 210Pbex measurements to investigate the sediment budget of a small forested catchment in Southern Italy. Hydrological Processes 27(6), 795-806.

[Figure]

**Fig. 1.** Fig. 1 (modified)

[Figure]

| Profile 1 | Profile 2 | Profile 3 | | | | |
|---|---|---|---|---|---|---|
| $^{137}$Cs (Bq kg$^{-1}$) | $^{137}$Cs (Bq kg$^{-1}$) | $^{137}$Cs (Bq kg$^{-1}$) | Mass depth (kg m$^{-2}$) | μ | σ | CV (%) |
| 34.0 | 35.0 | 35.0 | 25.9 | 34.7 | 0.6 | 1.7 |
| 30.0 | 18.0 | 25.0 | 40.1 | 24.3 | 6.0 | 24.8 |
| 17.6 | 26.0 | 22.0 | 50.7 | 21.9 | 4.2 | 19.2 |
| 19.4 | 22.0 | 9.0 | 66.3 | 16.8 | 6.9 | 40.9 |
| 10.0 | 11.0 | 19.0 | 79.0 | 13.3 | 4.9 | 37.0 |
| 9.3 | 15.0 | 7.0 | 89.2 | 10.4 | 4.1 | 39.4 |
| 10.2 | 6.0 | 13.0 | 100.0 | 9.7 | 3.5 | 36.2 |
| 9.3 | 7.4 | 5.1 | 115.8 | 7.3 | 2.1 | 29.1 |
| 5.0 | 6.1 | 10.0 | 138.2 | 7.0 | 2.6 | 37.4 |
| 4.2 | 4.0 | 4.0 | 167.4 | 4.1 | 0.1 | 3.2 |

**Fig. 2.** Figure (my example)

EXPONENTIAL PROFILE

[Figure]

PROFILE WITH MIGRATION PEAK

[Figure]

**Fig. 3.** Fig. 3 (modified)

---

## Referee Comment (RC2) · Anonymous Referee #2 · 13 Jan 2017

The paper addressed a very important topic in the soil erosion evaluation using Cs-137 technique. A good orientation about the reference site choice will define a different history about the soil erosion and deposition rates in the end. In my opinion the MS give us a good understanding about this and the complexity associate. In my opinion, the discussion about the commitment about the reference site is meaningful to the scientific community to reveal the uncertainties about it and to help to establish a protocol. However, I think the protocol will be site specific.

In general terms, it is quite difficult to establish a protocol to choose the reference value. I agree with the arguments and factors explored by the authors, but I'm not sure if the protocol suggested is the main contribution of the MS or even the application of this with the study of case showed. In my point of view the main contribution is the discussion about the control factor and the uncertainty associated. Maybe the paper

should be written more theoretical and with less pretension to establish a protocol applicable worldwide or to prove its application.

Another question/doubt is about the temporal variation. It was not so clear if the authors highly recommend a temporal analysis or no. If yes, how much time it takes to researchers decide if this is a good site to be used. Is this a constrain about the methodology proposed? Maybe the author could explore the fact the temporal evaluation will take a long time and maybe people will not be able to test it.

In my point of view the spatial variation is more pertinent and easy to be applicable. In the abstract the temporal variability is highlighted, for example, maybe the analysis could start with spatial variability and after the authors could show some insights about the temporal analysis.

Figure 1: Because this MS is proposing a reflexing/protocol, in my opinion the reference site should be chosen in a flat area in the top instead in the base of hillslope, for example, in a plateau without erosion/deposition possibilities.

Beside this, Maybe we can come back to the form and structure after the discussion about the points presented above.

Best regards

---

## Author Comment (AC1) · 31 Mar 2017

Dear Editor,

We received two very positive reviews with some excellent suggestions to improve the manuscript and the CheSS method giving it a more general relevance, since the suitability of references site is crucial, maybe even the most crucial step, in all FRN based erosion assessments.

The reviewers identified three main concerns regarding the proposed CheSS method. The first major point is that the approach might be site specific and not of general applicability. As respond to this point we included the spatial variability in a modified version of the decision tree (Figure 2) as such CheSS will be also applicable without temporal replicates. The second point was related to the time lag needed for the resampling approach. In general time span should be of sufficient length to cause an inventory change that it larger than the uncertainty related to the inventory assessment and the small scale variability e.g. larger 35%. We added information on this point in the ms. Moreover, we modified CheSS in a way that it is not mandatory to have temporal replicates. The final point was related to the criteria of the variability of the FRN depth profile. With the help of the example data of reviewer to we could establish a less strict data driven criteria, which in addition also allows to assess erosion and deposition processes (Figure 3).

With the updated presented manuscript, we took care to answer the reviewer's questions and comments as well as your editorial request. We hope you will consider our new improved submission for publication. Please do not hesitate if you have any further questions or concerns.

Looking forward to hearing from you.

Yours sincerely, Dr. Katrin Meusburger on behalf of Dr. Laura Arata and the co-authors

Reply Anonymous Referee #1 "In this paper the authors explored an important issue in the application of the 137Cs technique that relates to the choice of a reference site. They proposed a decision support tool (CheSS) to check the suitability of a reference site using repeated measurements of 137Cs undertaken in 2013 and 2015 on the same sites and measurements of 239+240Pu carried out in 2015. The basic assumption is that suitable reference sites are expected to present no significant temporal variation in their decay corrected 137Cs depth profiles. The authors individuated four main causes of possible variation in the inventory. These are (1) small scale variability connected to the non-homogeneus fallout (see areas affected by Chernobyl), (2) signs of erosion and/or deposition, (3) artefacts due to sampling and measurements, (4) turbation processes. The authors screened their six reference sites in the Urseren Valley (Switzerland) based on this assumption and tried to individuate the suitable reference site. The paper seems to me very in-teresting and I think it should be published but

some more details need to be added and/or discussed in this version.

Specific comments Introduction – The authors explain briefly the basic assumptions of the 137Cs with the help of Fig. 1. I agree with the explainations reported in the text but I think Figure 1 is a little bit misleading for people that are not familiar with the technique. In fact, what they depict as ‘reference site (R)’ is a valley bottom and, as is, it can be a depositional site. Also, what they depict as ‘depositional site (D)’ is a foot-slope and it may not necessarily be a ‘depression’ where deposition occurs. I suggest to redraw this figure in a more proper way (see my example below) Fig. 1 (modified)"

Reply 1_1: Thanks you for this comment. The depiction of this Figure was clearly driven by a site-specific adaptation of the FRN-method to our alpine sites. In our area, we cannot sample ridges, since they are very elevated and consist of bare rock. So we selected a ridge of a moraine at the valley floor, which for simplification was not depicted in the original Figure 1. We agree that the modified version of reviewer 1 will be of more general applicability and will change Figure 1 as suggested.

2.1 Repeated sampling strategy and calculation of inventories "This is a very important point. More work must be done to establish how long the time period between two sampling campaigns should be. This depends on the 137Cs inventory of the reference site and on its spatial variability. If the inventory is low, it is difficult to understand if the difference between the two sampling campaigns should be attributed to decay or to erosion and/or deposition or to the detector efficiency. In this case, a period of at least 10-15 years could be necessary. If the inventory is high and it is affected by the Chernobyl fallout, we could expect that the small scale spatial variability and the temporal variability are of the same order of magnitude and it is difficult to distinguish the relative contributions. Something like that is suggested by the data that the authors show in their Fig. 6. The inventory provided in 2015 are all higher than those obtained two years before (in 2013) with the only exception of Ref 6. This is not unexpected because it is not possible to relocate exactly the same sampling points. Clearly, more samples are necessary in this case. But also, a time period of two years between

two sampling campaigns could not be enough. The authors can add some comments here."

Reply 1_2: We will add some more discussion on the time lag needed between the repeated sampling. For sure there is no general advice and it will depend on the type of disturbance. An anthropogenic or animal disturbance can cause an immediate and significant change of the inventory. If the applicant also seeks to identify significant changes of the inventory due to erosion or deposition the time span should be of sufficient length to cause an inventory change that it larger than the uncertainty related to the inventory assessment e.g. larger 35%. For instance, in our case we have 700 Bq/m2 in the upper 3cm. We would need to loss or gain one third (1cm) of the soil to induce a significant change of the inventory. Assuming a bulk density of 0.5 g cm-3 this would correspond to 25 t ha-1yr-1 for the selected time lag of two years. We added a discussion on that point in line 98-101.

Node 2: No significant temporal variation of the 137Cs depth profile "I agree with the test related to the total inventory as explained in Node 1. However, I found node 2 too severe. I agree that the shape of the reference profile is important but, I think the test should be done on the entire profile not on the single layers. In many years of experience, I have never seen two profiles collected in the same site being identical. Maybe a practicle example can clarify my thoughts. Below there are 3 potential reference profiles characterised by the same total inventory (2510 Bq m-2), so they passed Node 1. They can be fitted by the same exponential model (same shape parameter h0 = 70 kg m-2 and same surface concentration A0 = 35 Bq kg-1). The values of cesium activity and mass depth for each layer are reported below. Figure (My example) Using the Sutherland range as a test (see suggestions in Node 2), in 5 cases out of 10 (see my values in red) the CV is greater than 35% (which is the upper limit suggested by Sutherland). If I have understood well, this result would suggest that the site where these profiles have been obtained is not suitable as reference site. I do not think I can agree with that because they show the same exponential decline with depth, and

the difference between each single layer can be attributed to other factors (the authors mentioned some of the other causes later in the paper). On the contrary, if we use a t-test or other statistical tests to compare mean and variance of these three profiles, I may be wrong, but I did not find any statistical difference. I think the authors should think about it and add some comments. There is another limitation in the application of this procedure suggested by the authors. In my example I have considered the same value of mass depth increment for the 3 profiles. This is an ideal case. In reality, due to differences in soil type, land use, presence of stones etc., it is difficult to obtain equal values of mass depth for the corresponding layers of different profiles. This makes this comparison not possible. In the end, I find more useful to check the shape of the entire profile."

Reply 1_3: Since the shape of the FRN depth profile is decisive in many conversion models, we consider it essential to retain a node to evaluate the shape in CheSS. However, we agree with reviewer 1 that the chosen criteria was too selective. With the help of the valuable example data supplied by the reviewer we modified the criteria of node 2. Instead of using a threshold value of the CV, we suggest to plot the depth profile of t0 against t1. If the shape of the profiles remained over time, the regression between the two depth profiles should follow a 1:1 line and the R2 >0.5. Taking into account that part of the variability may be explained by small scale variability of soil properties and adsorbed FRNs as well as procedural imprecision.

"Node 4 - Signs of disturbance associated with erosion and deposition processes I agree with the explaination in the text but, Figure 3 shows only the case where the 137Cs profile is perfectly exponential. In many places, profiles obtained in reference sites show a peak below the surface due to migration processes downward. In this case, the profile shown in figure 3b can be an undisturbed reference profile and, consequently, sheet erosion and deposition processes modify the shape accordingly. I suggest to improve their figure 3 considering both the possible cases (see my example below)." Fig. 3 (modified)

Reply 1_4: We agree with the concern of reviewer 1 in case of an none exponential depth distribution and thought about a data based criteria in addition to visual assessment to identify erosion or deposition processes between the two time steps. The regression equation established for the modified node 2 offers such a databased decision support. In case of accumulation the gradient of the trend line will be larger 1.1 and in case of erosion <0.9. We replaced Figure 3 with a new graph displaying this data driven approach (Figure 3, line 165). Please find below how your example data performs for the modified criteria of node 2&4:

"Page 7 – Line 21-22 – The authors say 'If information on the depth distribution of another FRN is available, this might provide a reliable confirmation'. I agree with this statement, but an example is necessary. The use of 210Pbex proved to be very effective in combination with or in alternative to 137Cs. In fact, good relationships exist between the results obtained with 137Cs and 210Pbex. I am sure the authors want to add some comments here maybe recalling some of the works done in this field (see for example Porto et al., 2006; 2013). Porto P., Walling D.E., Callegari G. and Catona F. (2006). Using fallout lead-210 measurements to estimate soil erosion in three small catchments in Southern Italy. Water, air and soil pollution: focus 6, 657-667 Porto P., Walling D.E., Callegari G. (2013). Using 137Cs and 210Pbex measurements to investigate the sediment budget of a small forested catchment in Southern Italy. Hydrological Processes 27(6), 795-806."

Reply 1_5: Indeed this is another good example, how other FRNs can underpin the selection of a suitable reference site. We included further discussion and references in line 214. Č Reply Referee 2 "The paper addressed a very important topic in the soil erosion evaluation using Cs-137 technique. A good orientation about the reference site choice will define a different history about the soil erosion and deposition rates in the end. In my opinion the MS give us a good understanding about this and the complexity associate. In my opinion, the discussion about the commitment about the reference site is meaningful to the scientific community to reveal the uncertainties about it and to

help to establish a protocol. However, I think the protocol will be site specific. In general terms, it is quite difficult to establish a protocol to choose the reference value. I agree with the arguments and factors explored by the authors, but I'm not sure if the protocol suggested is the main contribution of the MS or even the application of this with the study of case showed. In my point of view the main contribution is the discussion about the control factor and the uncertainty associated. Maybe the paper should be written more theoretical and with less pretension to establish a protocol applicable worldwide or to prove its application."

Reply 2_1: We can understand the concern of the reviewer, that a detailed protocol maybe too site-specific and may impede the application or adaptation of the reference site selection process for other areas. We will follow his/her suggestion in order to shift the focus of the ms away from a detailed protocol towards a theoretical concept. "Another question/doubt is about the temporal variation. It was not so clear if the authors highly recommend a temporal analysis or no. If yes, how much time it takes to researchers decide if this is a good site to be used. Is this a constrain about the methodology proposed? Maybe the author could explore the fact the temporal evaluation will take a long time and maybe people will not be able to test it."

Reply 2_2: So far, we did not provide a specific suggestion when temporal reference analysis could be beneficial and how long the time lag between the sampling should be. The time lag, as detailed in the reply R1_2 to reviewer 1, will depend on various conditions e.g. spatial variation, measurement uncertainty. In answer to the concern of reviewer 2 we will be more specific on this point and added to the existing paper in line 98-101.

"In my point of view the spatial variation is more pertinent and easy to be applicable. In the abstract the temporal variability is highlighted, for example, maybe the analysis could start with spatial variability and after the authors could show some insights about the temporal analysis."

R2_3: Thanks for this very good idea. We implemented the spatial approach in the abstract and implemented the spatial approach in CheSS (Figure 2) to make the concept more general applicable.

"Figure 1: Because this MS is proposing a reflexing/protocol, in my opinion the reference site should be chosen in a flat area in the top instead in the base of hillslope, for example, in a plateau without erosion/deposition possibilities."

R2_4: As explained above modifed Figure 1, displaying the reference site in a flat part of the ridge.

"Beside this, Maybe we can come back to the form and structure after the discussion about the points presented above. Best regards"

R2_5: We are sorry not to have replied in time to this comment because of the maternity leave of the corresponding author.

Please also note the supplement to this comment:
http://www.soil-discuss.net/soil-2016-72/soil-2016-72-AC1-supplement.pdf

| Node2 | | |
|---|---|---|
| P1 | P2 | P3 |
| 34 | 35 | 35 |
| 30 | 18 | 25 |
| 17.6 | 26 | 22 |
| 19.4 | 22 | 9 |
| 10 | 11 | 19 |
| 9.3 | 15 | 7 |
| 10.2 | 6 | 13 |
| 9.3 | 7.4 | 5 |
| 5 | 6.1 | 10 |
| 4.2 | 4 | 4 |

| Node 4 | deposition | erosion |
|---|---|---|
| P1 | P2 | P3 |
| 34 | 27 | 22 |
| 30 | 33 | 9 |
| 17.6 | 35 | 19 |
| 19.4 | 18 | 7 |
| 10 | 26 | 13 |
| 9.3 | 22 | 5 |
| 10.2 | 11 | 10 |
| 9.3 | 15 | 4 |
| 5 | 6 | 1 |
| 4.2 | 7.4 | 1 |

[Figure]

**Fig. 1.** Modified criteria with sample data supplied by reviewer 1

---

## Author Response (AR2)

Dear Editor,

We would like to thank you for your time and support in reviewing our manuscript. We carefully considered all your comments to the text and corrected it accordingly.

Thanks to your precious help and the input of the reviewers, we are confident that this new updated and improved version of the manuscript could be ready for its publication.

Please do not hesitate to contact us if you have any further questions or concerns.

Looking forward to hearing from you.

Yours sincerely,

Dr. Katrin Meusburger and Dr. Laura Arata

[revised manuscript text omitted]

Further, the depth profiles of the three replicates at reference site 1 presented also significant differences. We then looked at the differences in the skeleton content of the three replicates (Figure 2, B). An ANOVA test showed a significant difference (p-value of 0.025), thus,which indicates that a difference in the presence of stones in the three soil cores might have affected the FRN depth distribution. In particular, a Tukey's HSD (Honest Significant Difference) Post-hoc pairwise comparison identified the replicate number 3 at REF1 as a potential outlier. To validate the suitability of REF1 as a reference site, more replicates should be collected and measured, in order to compare their $^{137}$Cs depth profiles to the those results obtained during the first sampling campaign. In summary, REF2, REF4 (before the construction works) seem appeared to be most suitable for $^{137}$Cs-based studies. Form As for sites REF3 and REF5, a visual inspection of their soil profiles excluded that

any soil disturbance affected the sites (Figure 2, C). Consequently, at those sites, the variation in their depth profiles is due to a heterogeneous fallout with high spatial variability (Figure 2, D). 
[revised manuscript text omitted]

Oh, J. S., Lee, S. H., Choi, J. K., Lee, J. M., Lee, K. B., & Park, T. S. (2014). Atmospheric input of 137 Cs and 239,240 Pu isotopes in Korea after the Fukushima nuclear power plant accident. *Applied Radiation and Isotopes, 87*, 53-56.

400 Owens, P.N., Walling, D.E., 1996. Spatial variability of caesium-137 inventories at reference sites: an example from two contrasting sites in England and Zimbabwe. Appl. Radiat. Isot. 47, 699–707.

Parsons, A. J. and I. D. L. Foster,2011. "What can we learn about soil erosion from the use of 137Cs?" Earth Science Reviews 108: 13.

Pennock, D.J., Appleby, P.G., 2002. Site selection and sampling design, in: Handbook for the Assessment of Soil Erosion
405 and Sedimentation Using Environmental Radionuclides. Springer, pp. 15–40.

Porto, P., Walling, D.E., Alewell, C., Callegari, G., Mabit, L., Mallimo, N., Meusburger, K., Zehringer, M., 2014. Use of a 137 Cs re-sampling technique to investigate temporal changes in soil erosion and sediment mobilisation for a small forested catchment in southern Italy. J. Environ. Radioact. 138, 137–148.

Porto P, Walling DE, Callegari G., 2013. Using $^{137}$Cs and $^{210}$Pb$_{ex}$ measurements to investigate the sediment budget of a small
410 forested catchment in southern Italy. Hydrological Processes; 27: 795-806.

Ritchie, J. C., Ritchie, C. A., 2001. Bibliography of publications of Cesium-137 studies related to erosion and sediment deposition. USDA-ARS Hydrology and Remote Sensing Laboratory.

Schaub, M., Alewell, C., 2009. Stable carbon isotopes as an indicator for soil degradation in an alpine environment (Urseren Valley, Switzerland). Rapid Commun. Mass Spectrom. 23, 1499–1507.

415 Sutherland, R.A., 1996. Caesium-137 soil sampling and inventory variability in reference locations: A literature survey. Hydrol. Process. 10, 43–53.

Walling, D.E., He, Q., Appleby, P.G., 2002. Conversion models for use in soil-erosion, soil-redistribution and sedimentation investigations. In: Zapata, F. (Ed.) Handbook for the assessment of soil erosion and sedimentation using environmental radionuclides. Kluwer, Dordrecht. Netherlands. pp. 111–164.

420 Winiarek, V., Bocquet, M., Duhanyan, N., Roustan, Y., Saunier, O., & Mathieu, A. (2014). Estimation of the caesium-137 source term from the Fukushima Daiichi nuclear power plant using a consistent joint assimilation of air concentration and deposition observations. *Atmospheric environment, 82*, 268-279.

Zapata, F. (Ed.), 2002. Handbook for the assessment of soil erosion and sedimentation using environmental radionuclides (Vol. 219). Dordrecht: Kluwer Academic Publishers.

425 Zollinger, B., Alewell, C., Kneisel, C., Meusburger, K., Brandová, D., Kubik, P., Schaller M., Ketterer M., Egli, M., 2014. The effect of permafrost on time-split soil erosion using radionuclides ($^{137}$Cs, $^{239+240}$Pu, meteoric $^{10}$Be) and stable isotopes ($\delta$ $^{13}$C) in the eastern Swiss Alps. Journal of Soils and Sediments, 15(6), 1400–1419.